# Experimental Efficacy of the Face Shield and the Mask against Emitted and Potentially Received Particles

**DOI:** 10.3390/ijerph18041942

**Published:** 2021-02-17

**Authors:** Jean-Michel Wendling, Thibaut Fabacher, Philippe-Pierre Pébaÿ, Isabelle Cosperec, Michaël Rochoy

**Affiliations:** 1Occupational Health and Safety, ACST, F-67000 Strasbourg, France; jean-michel.wendling@acst-strasbourg.com; 2Department of Public Health, GMRC, CHRU, F-67000 Strasbourg, France; thibaut.fabacher@chru-strasbourg.fr; 3NexGen Analytics, Sheridan, WY 82801, USA; philippe.pebay@ng-analytics.com; 4PharmD, F-94210 St Maur-des-Fossés, France; isacoco67@msn.com; 5General Medicine Department, University Lille, CERIM, ULR 2694, F-59000 Lille, France

**Keywords:** face shields, masks, particles, aerosolization, covid19, covid, coronavirus infections, aerosols, pneumonia, prevention and control, protective devices, pandemics, emitter, receiver, experimental setup

## Abstract

There is currently not sufficient evidence to support the effectiveness of face shields for source control. In order to evaluate the comparative barrier performance effect of face masks and face shields, we used an aerosol generator and a particle counter to evaluate the performance of the various devices in comparable situations. We tested different configurations in an experimental setup with manikin heads wearing masks (surgical type I), face shields (22.5 cm high with overhang under the chin of 7 cm and circumference of 35 cm) on an emitter or a receiver manikin head, or both. The manikins were face to face, 25 cm apart, with an intense particle emission (52.5 L/min) for 30 s. The particle counter calculated the total cumulative particles aspirated on a volume of 1.416 L In our experimental conditions, when the receiver alone wore a protection, the face shield was more effective (reduction factor = 54.8%), while reduction was lower with a mask (reduction factor = 21.8%) (*p* = 0.002). The wearing of a protective device by the emitter alone reduced the level of received particles by 96.8% for both the mask and face shield (*p* = NS). When both the emitter and receiver manikin heads wore a face shield, the protection allowed for better results in our experimental conditions: 98% reduction for the face shields versus 97.3% for the masks (*p* = 0.01). Face shields offered an even better barrier effect than the mask against small inhaled particles (<0.3 µm–0.3 to 0.5 µm–0.5 to 1 µm) in all configurations. Therefore, it would be interesting to include face shields as used in our experimental study as part of strategies to reduce transmission within the community setting.

## 1. Introduction

At the end of 2019, a novel coronavirus named Severe Acute Respiratory Syndrome Coronavirus 2 COVID (SARS-CoV-2) emerged [1]. The outbreak of the disease caused by this virus, Coronavirus Disease 2019 (COVID-19), was declared a pandemic by the World Health Organization (WHO) on 11 March 2020, and has caused nearly 1 million fatalities as of 27 September 2020 [2].

The airborne transmission route for SARS-CoV-2 is virulent for the spread of COVID-19 [3,4,5], as for SARS-CoV-1 [6]. At the present time, we have not identified the precise aerosol viral load or the minimum infectious dose of SARS-CoV-2 to cause an infection [7]. A viable virus can be emitted by an infected person by talking, singing, coughing or sneezing [8]: a small fraction of individuals are considered to be “speech super-emitters”, releasing more particles than others [9].

A challenge in pandemic control is limiting the transmission of SARS-CoV-2 by asymptomatic or pre-symptomatic individuals [10]. A systematic review of the literature and meta-analysis revealed that face covering decreased the risk of airborne infections [11]. Surgical face masks significantly reduced detection of coronavirus RNA in aerosols [12]. Face covering by asymptomatic people (the primary case and family contacts before this primary case had symptoms) is effective in reducing transmission [13]. In the United States, an analysis revealed that the difference with and without mandated face covering represented the principal determinant in shaping the trends of the pandemic [4].

However, in Western countries, there has been significant controversy over the face covering [14,15], notably after the recommendation of the WHO on 5 June 2020 [16]. To improve compliance and acceptance, face shields are a good compromise and have many advantages: they are much more acceptable to young children [17], preferable during intensive aerobic physical activity, or for people who are anxious about wearing a mask [18]. Face shields could also be of economic and ecological interest because they are washable and therefore reusable.

There was controversy surrounding a recommendation to wear a face shield during this period, because little is known about the efficacy of different types of protective measures in the context of this pandemic [19]. The Centers for Disease Control and Prevention (CDC) “does not currently recommend” the use of face shields as a substitute for masks because “there is currently not enough evidence to support the effectiveness of face shields for source control” [20].

It has already been shown that the use of face shields can significantly reduce healthcare workers’ short-term exposure to large infectious aerosol particles. Some authors think they are less effective against small particles (size less than 5 microns), which could remain suspended in the air for long periods of time and could easily get in through the wide holes on the sides and at the bottom and be inhaled. The space between the face and the face shield is indeed larger than that of the mask [21].

We hypothesized that face shields, worn by the emitter and worn by the receiver, could reduce the amount of particles from <0.3 µm to 10 µm emitted and received. We aimed to quantify the number of particles of different sizes detected in the mouth of a manikin head with different types of protections (face shield, mask or nothing); and to compare these different types of protection in different situations.

## 2. Materials and Methods

### 2.1. Experimental Setup

The evaluation was carried out in August 2020 on an experimental setup with two manikin heads positioned at 1.70 m high and at 25 cm from each other (Figure 1). We opted for a short distance of 25 cm in order to limit the background noise and to place ourselves in a short but intense exposure condition. This situation may exist in real-life situations, particularly in some public transport systems. The tests were carried out in an empty closed room of 18.40 m^2^ (and height of 2.5 m, i.e., a volume of 46 m^3^) without drafts or mechanical ventilation, and with a sectional door of size 8.4 m^2^ closed during the tests.

One of the two heads (called an “emitter” or Em) has been hollowed out to reproduce a mouth. A pipe was introduced through a hole in the head to connect the fogger to the mouth. The atomizer was generated by aerosolizing distilled water with a fogger TRIXIE Fogger XL.

The other head, called “receiver” (Re) has also been hollowed out at the mouth. A short pipe of 5 cm long and 15 mm internal diameter was connected to an optical particles counter.

The optical particles counter PC220 (TROTEC, Hamsberg, Germany) is designed to measure the size and the number of particles in the air. It sucks a volume of air for an adjustable amount of time and determines the size and amount of particles contained in it. The device is equipped with an integrated measuring cell with a laser (class 3R laser, 780 nm, 1.5–3 mW). According to ISO 21501, the counting efficiency is 50% at 0.3 µm and 100% over 0.45 µm.

We generate aerosols with pure water. Particles of sizes less than 0.3 µm, 0.3 µm to 0.5 µm, 0.5 µm to 1 µm, 1 µm to 2.5 µm, 2.5 µm to 5 µm, and 5 µm to 10 µm were treated equally during the process. The cumulative counting method was performed for the analysis. The amount of all particles up to the selected particle sizes were counted (e.g.: “0.5 µm = 417” means that 417 particles had a size between 0.3 and 0.5 µm). The pumping time, air volume and the start delay are programmable. A HEPA filter (TROTEC, Hamsberg, Germany) was used on the counter to reset to zero before each measurement.

The temperature, ambient humidity and the aerosol flow velocity were recorded using a hot-wire thermo-anemometer TA300 (TROTEC, Hamsberg, Germany). This device comes equipped with a hot-wire sensor and microprocessor technology for signal amplification. This combination guarantees precise measuring results.

Two types of personal protective equipment (PPE) were tested and their barrier performances were compared (Figure 2): EN14683 surgical masks type I (over 95% of 3 µm particle filtration) (COVEIX, La Chatre-Indre, France); face shields (Viseira CDX— CODIL^®^, Fajöes, Portugal) covering the eyes, mouth, nose, 22.5 cm high with overhang under the chin of 7 cm, circumference of the visor 35 cm, front opening 4 cm high in line with the center of the forehead.

### 2.2. Description and Analysis of Different Test Setups

The background level of particulate pollution was first evaluated in the experimental room before (situation 0 with 10 measurements) and between each experiment (Figure 3: situation 1–4a).

The aerosol was then generated, without any PPE. The amount of particles and the size distribution of the aerosol were measured at a distance of 25 cm (10 measurements) (situation 1). This configuration is the reference configuration with the maximum exposure of the receiver without any protection to which all the others will be compared.

Six configurations with PPE devices were then tested each time with a series of 10 measurements (Figure 3): surgical mask (situation 2), then face shield (situation 2a) on the receiver head only; surgical mask (situation 3) then face shield (situation 3a) on the emitter head only; surgical mask (situation 4) then face shield (situation 4a) on both emitter and receiver heads (after named “double protection”).

The aerosol air flow coming out of the generator was continuous at 52.5 l/mn. The speed airflow mean value was 4.95 m/s (sd = 0.17 m/s) (*n* = 20). The airflow speed of our generator was representative of the exhaled air velocity when talking [22]. The counter was started with a 5 s programmed delay from the generator, time to ensure that the head of the receiving manikin was well surrounded by the particle flow. The particle counter calculated the total cumulative particles aspirated on a volume of 1.416 L. Counting was performed on the 6 channels (<0.3 µm–0.3 to 0.5 µm–0.5 to 1 µm–1 to 2.5 µm–2.5 to 5 µm and 5 to 10 µm) during 30 s of sucking in air. After each measurement, the counter was reset to zero by the HEPA filter and the room ventilated by opening the sectional door for at least 5 min to remove airborne particles. Before and between testing, the background particles level, temperature and humidity were recorded. Our Emitter–Receiver basic configuration was not changed during the entire experimentation: only masks and face shields have been exchanged for the data acquisition. First, the amount of particles and the distribution of the particle size were studied without any PPE.

For the experiments performed with the aerosol particle measurement instruments, the parameters studied and compared were: the impact of the presence of PPE, the difference between protection of the emitter, receiver or both, the influence of each type of PPE, mask or face shield and the influence of the aerosol particle sizes on the results (<0.3 µm–0.3 to 0.5 µm–0.5 to 1 µm–1 to 2.5 µm–2.5 to 5 µm and 5 to 10 µm, respectively).

### 2.3. Statistical Analysis

We evaluated and compared the barrier performance of each device in 6 different configurations by the Reduction Factor (RF) of the particles received and inhaled by the manikin head according to the following formula (Equation (1)):(1)$$\mathrm{RF}\;=\frac{\mathrm{Particles}\;\mathrm{received}\;\mathrm{without}\;\mathrm{any}\;\mathrm{PPE}\;-\;\mathrm{Particles}\;\mathrm{received}\;\mathrm{with}\;\mathrm{the}\;\mathrm{tested}\;\mathrm{PPE}}{\mathrm{Particles}\;\mathrm{received}\;\mathrm{without}\;\mathrm{any}\;\mathrm{PPE}}$$

The percentage of total inhaled particles (PTIP) is obtained by the following formula (Equation (2)):PTIP = 1 − RF(2)

We also present the results of the barrier effect of each device (mask or face shield) using a particle size range approach.

For descriptive analysis, we used mean value, median and standard deviation. To compare the reduction factors between the mask and face shield and the impact of location (emitter vs. receiver) of each protection, we performed a nonparametric Mann–Whitney Wilcoxon test (with *p*-value computed for each comparison). We note that the independence assumption holds because the shield-group and the mask-group observations were successively measured in different runs of the same experimental setting. The alpha risk was set at 5% for all analyses.

## 3. Results

### 3.1. Environmental Factors Measurements

The temperature and hygrometry in the experimental room were regularly measured (*n* = 20). The mean values were respectively 27.73 °C (sd = 0.50 °C) and 68.3% for relative humidity (sd = 2%). Before aerosolization, the thermo-anemometer confirmed the absence of significant air current in the room around the test bench (values <0.05 m/s in all directions). During the production of aerosol by the fogger, in the first step of the study, the speed of the airflow coming out of the pipe without any protection was evaluated at the mouth of the manikin head with the thermo-anemometer.

### 3.2. Aerosol Total Particle Measurements

The background level of particulate pollution on the sucked-in air volume standard in the experimental room showed a mean value of 13,837 particles (sd = 2436), i.e., 9.77 particles/cm^3^ (Table 1). The background aerosol level was checked between each experiment and was between 11,383 and 17,955 particles.

The particles aerosol amount, received at 25 cm, decreased according to size. The average amount of particles measured at 25 cm without protection was 1,000,763 particles (sd = 165,118), i.e., 707 particles/cm^3^. Of all the particles, 60% were less than 0.3 µm and 94% less than 1 µm (Figure 4). The number of detected particles decreased with increasing particle sizes.

### 3.3. Efficacy of Face Covering: Total Particle Reduction Factor Depending on the Situation

We evaluated the quantity of particles that were stopped by the different devices and calculated reduction factors (RF) and percentages of total inhaled particles (PTIP) in each of the 6 configurations. Thus, the reduction factors were compared.

In our experimental configuration, when the receiver head wore a mask, the RF was 21.8%; when the receiver head wore a face shield, the RF was significantly higher: 54.8% (*p* = 0.002). When the emitter head wore a mask, or a face shield, the RF was similar: 96.7% (no difference). When the receiver and the emitter wore a mask, the RF was 97.3%; when the two wore a face shield, the RF was significantly higher: 98.0% (*p* = 0.011) (Table 1, Figure 5). 

### 3.4. Efficacy of Face Covering Depending on Particle Size Range

Considering all particle sizes (≤0.3 µm to 10 µm), the reduction factors were always better or similar with the face shield compared to the mask. The face shield, when worn only by the receiver, was always more effective in blocking particles than the mask, and for all particle ranges. The face shield performed significantly better when the particles were smaller: RF were respectively in the range of <0.3 µm: 47.9% versus 13.8% for mask (*p* = 0.006), in the range of 0.3–0.5 µm: 47.6% versus 6.14% for mask (*p* = 0.02), in the range of 0.5–1 µm: 81.9% versus 60.9% for mask (*p* = 0.009), in the range of 1–2.5 µm: 92.3% versus 79.8% for mask (*p* = 7.5.10^−5^) in the range of 2.5–5 µm: 97.1% versus 91.9% for mask (*p* = 0.009), in the range of 5–10 µm: 99.6% versus 98.5% for mask (*p* = 0.0006) (Figure 6, Table A1).

The face shield, when worn only by the emitter, performed about the same as the mask for all sizes of particles with more than 96% reduction factor for the particles less than 1 µm and 99% reduction factor for the particles more than 1 µm in size (Figure 7, Table A1).

When double protection was worn (i.e., mask for Re and Em or face shield for Re and Em), the face shield performed significantly better with a reduction factor of 97.7% versus 96.8% for the mask in the range of <0.3 µm (*p* = 0.01), 97.8% versus 97.1% for the mask in the range of 0.3–0.5 µm (*p* = 0.052 NS). For particles more than 1 µm in size, masks and face shields worn by the emitter and the receiver reduced the amount of inhaled particles by more than 99% (Figure 8, Table A1).

### 3.5. Comparison between Emitter or Receiver Protection

Statistically fewer particles are detected when the protection is worn by the emitter alone (RF = 96.7%) compared to the protection worn by the receiver alone (*p* = 10^−5^) (Table 1).

## 4. Discussion

Few studies have evaluated the benefits of face shields in limiting infectious transmission (speaking, singing, sneezing, coughing, etc.) when worn by infected persons, whether symptomatic or not. We experienced a short exposure (30 s) at a close distance (25 cm) with an intense airflow (52.5 L/mn at a mean speed value of 4.95 m/s) in an enclosed unventilated space of 46 m^3^. The particle counter calculated the total cumulative particles aspirated on a volume of 1.416 L.

In our experimental situation, face shields and masks could reduce the amount of particles emitted and finally received by the target. The number of inhaled particles detected on the receiver decreased mainly when the emitter wore face protection, and more so when both the emitter and receiver wore face protection. Our results indicate a better outcome when the emitter wears face protection (altruistic port) [15,23].

As part of the fight against the spread of SARS-CoV-2, face masks have been generalized to prevent propagation from asymptomatic emitters, as effective “anti-droplet screens” [23,24,25]. Another at-risk situation for SARS-CoV-2 transmission may be through the hands: indeed, SARS-CoV-2 can survive up to 9 h on human skin, much longer than the influenza virus, and is found in wastewater and stools [26,27,28]. Surgical masks may be less protective for this pathway than face shields that prevent finger-to-eye contamination in addition to hand-to-mouth; however, current evidence does not suggest that this pathway is predominant [5].

The current hypothesis is that particles emitted when speaking or breathing are mainly formed by a “fluid-film bursting” mechanism inside the small airways of the lungs and/or by the vibration of vocal folds in the larynx [4,29]. The amount of aerosols emitted during speech correlates with the loudness of vocalization, physiological factors and language spoken, ranging from approximately 1 to 50 particles per second, with some “super-emitters” which release much more particles in quantity than their peers. Particle emission when speaking shows a peak size of 0.75 to 1 µm and maximum size around 7 µm [9,22,30,31]. The particles smaller than 1 µm were easily reproduced by our aerosol generator.

Under our experimental conditions, the total number of particles received was significantly lower when wearing a face shield than when wearing a mask by the receiver, especially for microparticles around and less than 1 µm, the ones which are emitted during speaking. Face shields are commonly used by healthcare workers to protect the face; the CDC strongly recommends to wear a face shield covering the front and sides of the face, a mask with an attached shield, or a mask and goggles, during aerosol-generating procedures on patients not infected with M. tuberculosis, SARS, hemorrhagic fever viruses or other viruses requiring the use of N95-type protections [32].

Lindsley et al. studied face shields and concluded that they can reduce healthcare workers’ short-term exposure to large particles of infectious aerosols. The effectiveness of the face shield, reducing the amount of inhalation exposure to influenza virus on the receiver, was estimated to range between 68% and 96%. After 30 min, the effectiveness was 80% and face shields stopped 68% of aerosols [21]. One recent pre-print study found that the face shield blocked more than 90% of the otherwise inhaled particles; for finer particles (0.3 μm), the face shield performed much better, as in our study [33].

If a transmitter wears a mask or face shield, the source control of inter-individual transmission is expected to improve [18,34]. Surgical masks are now the gold standard in the fight against COVID-19 to control the source emission. There are some limitations with the surgical mask, including potential permeability to particles less than 3 microns, leaks even if the filtration performance is announced at more than 95%, often improper wear and variable acceptability. Despite these limitations, its effectiveness has been widely demonstrated [12,14,16,20,23,24,35,36]. To our knowledge, few studies have compared the ability of source control between face shields and masks in an identical configuration or in a dual protection (emitter and receiver). Face shields were often used according to the paradigm of personal protection, as before the pandemic in Western countries [15].

Our results differ from those of Verma et al.: to evaluate the performance of the face shield as source control, they used a cough simulator, synthetic smoke and two lasers (horizontal and vertical); by placing a plastic face shield, they found that smoke particles spread behind the emitter [37]. However, they did not quantify the number and the distribution of particles emitted, or the decreased concentration with distance. In addition, they did not use an aerosol consisting of water-based liquid particles, but a smoke generated at high temperature, which behaves differently. The face screen was positioned semi-open, in an improper way, facilitating the exit of a plume of exhaled air; in contrast to Verma et al., we used a liquid aerosol generator mimicking the particles emitted by the voice at room temperature, with a right-angle position close to the chin down, a chin and forehead overhang, and most importantly, a position of the visor parallel to the face [37].

Some other experimental studies used collection chambers with a manikin head placed in direct contact with the walls and in very small enclosed spaces, sometimes with an air suction device [38,39]. Our experimental conditions are closer to a real-life situation: we take into account the effect of air dilution in an open space with heads located at the height of standing, average-sized individuals. However, these results are also in line with those showing that mask protection is never completely effective in small, poorly ventilated enclosed spaces; moreover, the wearing of facial protection must be always associated with respect for physical distancing [36,40].

In another experimental pre-print study [41], researchers used Background-Oriented Schlieren (BOS), imaging to compare several types of face protections (surgical mask, homemade mask, commercial face shield and a 3D-printed face shield). The results showed that during quiet, heavy breathing or coughing, no front throughflow was discernible for the lightweight 3D-printed face shield. For surgical masks, exhaled air travelled more than 23.7 cm. The maximum distances travelled by the air flows (crown, brow, side and back jets) were all greater for the surgical mask compared to the face shield. Consistent with our observations, face shields generate particularly strong air flows towards the ground while no flow was discernible with the surgical mask in that direction: of course, it seems more desirable to direct the flow of infected particles towards the ground rather than letting them escape in the direction of the receiver.

Arumuru et al. [42] compared the “barrier” performances against saliva spray emitted on different devices using a camera and laser illumination. During a sneeze projecting aerosols at 25 feet, the distance was respectively reduced to 2.5 feet with the polypropylene surgical mask (−90%), 1.5 feet with the double-layer cotton mask (−94%), 1.5 feet with the triple-layer cotton mask (−94%) and only 1 foot with a polycarbonate face shield (−96%). The study shows that a face shield had the best performance of all the devices and that the surgical mask is the one that provides the weakest barrier to sneezing. With an N95 mask, the flow has been visualized at a distance of 2 feet behind the manikin [42]. So, face shields do seem to be a good alternative to protecting others in experimental studies.

Face shields are commonly used as an infection control alternative to goggles, because they also protect other areas of the face (forehead, preauricular area, cheeks, chin, etc.) and limit splashes from the face especially to the eyes.

A significant consideration is that face shields reduce the potential for contamination through the eyes. This property is particularly interesting in the fight against COVID-19. This route is rarely studied and the eye protection could be a determinant in community settings [11,43]. A study performed in Hubei hospital found that only 5.8% of COVID-19 patients (16 of 276 patients) wore glasses compared to an estimated 31.5% of the general population, suggesting that eye protection, eyeglasses or face shields could be useful against COVID-19 [43]. The ocular mucosa and the nasopharynx are connected by the nasolacrimal duct. When splashes reach the cornea or conjunctiva, they can penetrate the nasolacrimal duct and be transported to the nasopharynx and trachea [44,45]. Ocular manifestations seem frequent, and high-frequency hand–eye contact correlates with conjunctival congestion [46,47]. Immunohistochemical analysis also revealed the expression of ACE2 and TMPRSS2 in the conjunctiva, the limbus and the cornea, which suggests that cells on the ocular surface are susceptible to SARS-CoV-2 and could therefore serve as a gateway and reservoir for interindividual transmission [44,48,49]. In a meta-analysis, eye protection was associated with a lower risk of infection in 13 studies (RR = 0.34) and 2 adjusted studies (aOR = 0.22). Eye protection alone reduced the infection risk by 78% (aOR = 0.22) [11].

In addition to eye protection, face shields offer a number of advantages [18,19]. They are easily reusable and washable with soap or antiseptic [50]. The economic interest is obvious for the poorest populations throughout the world. They prevent the wearer from touching their mouth and nose. The face shield does not need to be constantly readjusted like the mask. The face shield, if its external surface is contaminated by the hands, cannot be a source of external contamination, unlike the mask, because the screen is watertight. The face shield is better tolerated during intense physical effort. It seems to limit the appearance of fog on eyeglasses compared to masks and thus could reduce the risk of accidents that occur while driving. They should be better tolerated in children, for whom the importance of facial protection is increasingly demonstrated in the fight against the population spread of COVID-19 [51]. People who wear a medical mask sometimes take it off to communicate, for comfort or to facilitate lip-reading, especially when dealing with people who are hard of hearing [52]. The face shield makes it easy to visualize important facial expressions in emotion readings, which is a crucial part in communication.

However, our study has some limitations: the experimentation configuration is an extreme configuration of exposure. Considering human particles (p) emission concentrations (concentrations ranged from 2.4 to 5.2 p/cm^3^ for coughing and 0.004–0.223 p/cm^3^ for speaking) [30,53,54], our aerosol generator produced much more particles with a very short distance and short time of 30 s in an enclosed, unventilated space. Particles inhalation by aerosolization could also be achieved by long, low-intensity exposure: this is not tested here in our study. The study was only conducted on one type of surgical mask (type I) and one type of face shield model; other models could be more or less efficient.

A face shield could have a higher RF because the distance is very close: the particle barrier effect is more significant, especially for the case where the Em is wearing a shield. The particle counter calculated the total cumulative particles aspirated on a volume of 1.416 L; this volume can be a part of the volume behind the face shield (kind of reservoir of clean air for the receiver). If the screen were worn in a more realistic context by a human, the movement and prolonged breathing could likely quickly display the original “clean” air behind the mask (1.416 L is equivalent to about 2–3 breaths).

Real life leads to situations where the emitter is not always facing the receiver, but we made the ballistic assumption that the riskiest situation was face-to-face to compare surgical masks and face shields in the exact same configurations [9,30]. The way in which one wears a mask or face shield, in addition to the shape and design of these protections, can be a determining factor in the efficiency of its performance [41]. Further experimentations should confirm performance through the shape and the face shield design, and efficiency for long-term exposure. To protect others in the best possible way, the objective should be to send the exhaled air flow towards the wearer or towards the ground with the most efficient design of the visor. More studies are needed in real-life situations to determine the best possible uses and effectiveness of face shields. Some microbiological studies on sick people equipped with face shields with environmental surfaces analysis and on air bio collectors should be conducted. The role of droplet evaporation under experimental conditions (temperature, humidity) was not taken into consideration. New studies varying these parameters are also useful to better compare PPE. Finally, it could be relevant to study the effect of face shields over a longer period of time, and with a double individual protection combining the wearing of a mask and a face shield, particularly as a transmitter.

## 5. Conclusions

Our results show that, in our experimental conditions of short and face-to-face exposure, when the receiver alone wore a face shield, the amount of particles was significantly reduced compared to when the receiver alone wore a mask, even with small particle size emission (≤0.3 µm). When the emitter wore a face shield or mask, it was more efficient than when only the receiver wore protection, with no difference between mask or face shield. In our experimental conditions, the double protection (for Em and Re) allowed for even better results: 98% reduction for face shields and 97.3% for masks. Further studies are needed to evaluate experimental efficacy in the context of long-term exposure. The efficiency of the face shields in our study encourages the development of their use for short exposure situation, or where no facial protection is possible (school cafeterias, bars...) through the integration of large Plexiglas between users. Finally, the results of our experimental study suggest that well-covering face shields should be included as part of an expanded arsenal against SARS-CoV-2, as a possible face covering alternative for people with medical intolerance to masks, for situations where no facial protection is used (collective sporting or cultural practices without masks) or in addition to masks.

## Figures and Tables

**Figure 1 ijerph-18-01942-f001:**
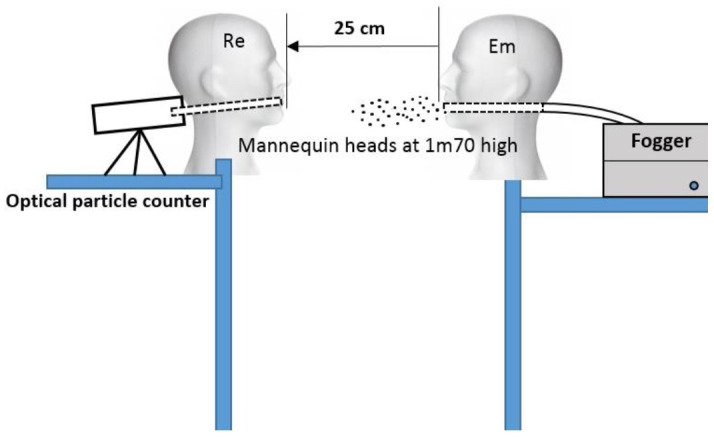
Experimental setup; Em: emitter; Re: receiver.

**Figure 2 ijerph-18-01942-f002:**
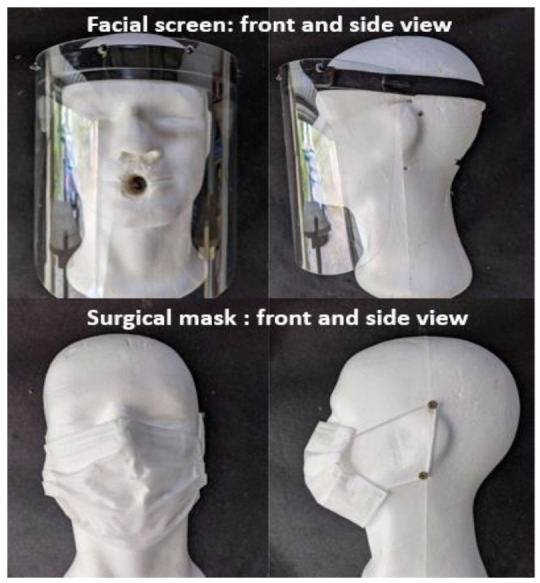
Surgical mask type I and face shield (front and profile views).

**Figure 3 ijerph-18-01942-f003:**
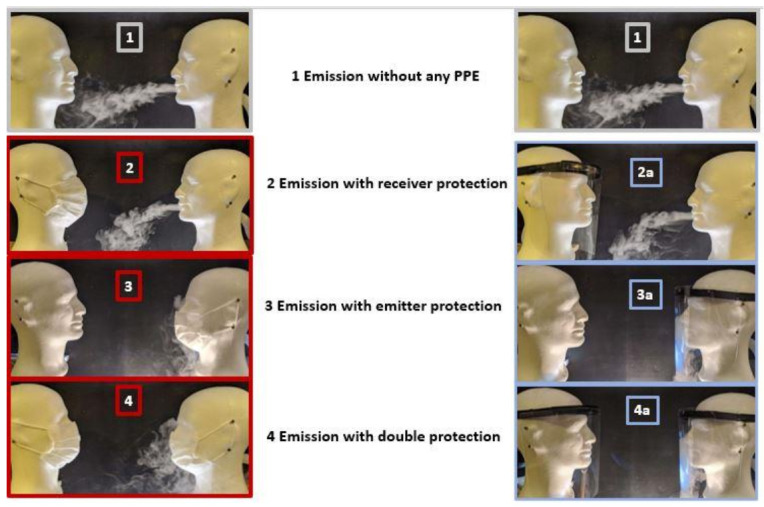
Experimental configurations: 1: emission without any personal protective equipment (PPE); 2: emission with mask for Receiver protection; 2a: emission with face shield for Receiver protection; 3: emission with mask for Emitter protection; 3a: emission with face shield for Emitter protection; 4: double protection with masks for Emitter and Receiver; 4a: double protection with face shields for Emitter and Receiver.

**Figure 4 ijerph-18-01942-f004:**
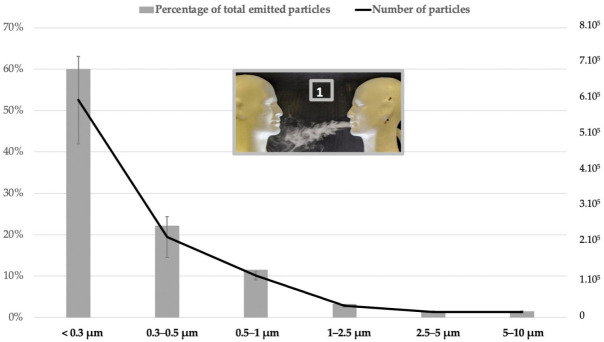
Particles distribution received at 25 cm without any PPE (situation 1).

**Figure 5 ijerph-18-01942-f005:**
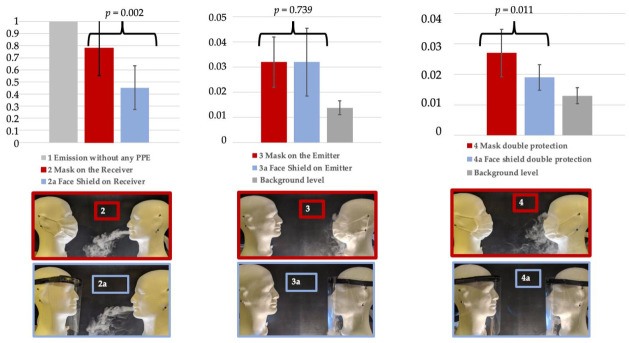
Reduction factors for the total particles: mask vs. face shield.

**Figure 6 ijerph-18-01942-f006:**
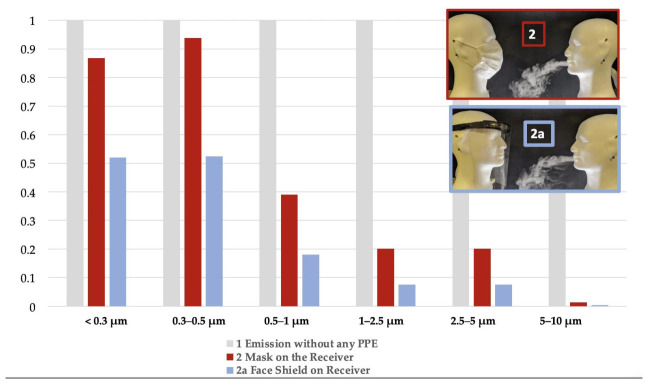
Percentage of Total Inhaled Particles (PTIP) with protection only for Receiver (Re): mask vs. face shield for each particle size range compared to lack of protection.

**Figure 7 ijerph-18-01942-f007:**
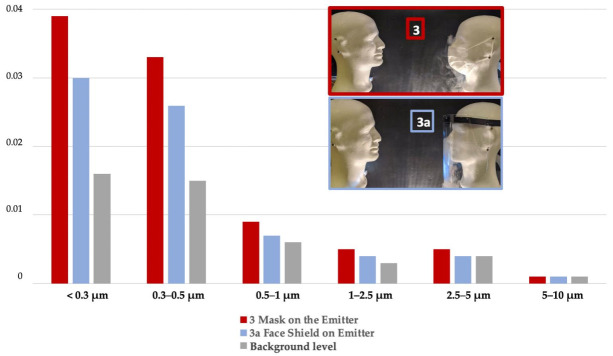
Percentage of Total Inhaled Particles (PTIP) with protection only for Emitter (Em): mask vs. face shield for each particle size range compared to lack of protection.

**Figure 8 ijerph-18-01942-f008:**
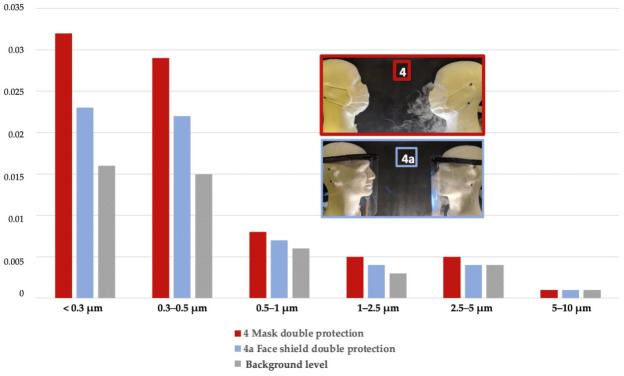
Percentage of Total Inhaled Particles (PTIP) with Double protection (Em+Re): mask vs. face shield for each particle size range compared to lack of protection.

**Table 1 ijerph-18-01942-t001:** Reduction factor (RF) and Percentage of Total Inhaled Particles (PTIP) in the different configurations.

Particle Size	Protection	Device	Mean Value (Total Number of Particles)	Standard Deviation	Median	Mean Value per cm^3^	Reduction Factor	*p* Value
Total of particles	Emission without any PPE		1,000,763	165,118	1,056,554	707	-	
	Receiver	mask	782,927	228,860	827,361	553	0.217	*p* = 0.002
		face shield	452,600	176,387	467,623	320	0.547	
	Emitter	mask	32,495	10,034	29,308	23	0.967	*p* = 0.739
		face shield	32,085	13,525	28,124	23	0.967	
	Double protection	mask	26,970	7818	26,044	19	0.973	*p* = 0.011
		face shield	19,707	4206	19,299	14	0.980	
	Background level		13,837	2436	13,730	10	-	

## Data Availability

Data available on request.

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
