# Peer review of "Experimental Efficacy of the Face Shield and the Mask against Emitted and Potentially Received Particles"

_ijerph, 2021, doi:10.3390/ijerph18041942_

Round 1
Reviewer 1 Report
The authors present results from an experiment aiming to quantify the ability of different forms of Personal Protective Equipment (PPE) to limit aerosol transmission between two individuals. The two pieces of PPE studied include face shields and masks. The experiments conducted aim to reproduce quick, but proximal and intense, periods of exchange between two individuals, similar to situations common in healthcare settings. The authors conclude that both masks and face shields are very effective at reducing emitted particles, while the face shield provided moderate protection to the receiving individual but more than masks which provided only very limited protection when worn by the receiver.
The work is obviously quite relevant given the present CoVID-19 pandemic and the subject matter is certainly appropriate for IJERPH. The manuscript is well organized and clear and the methodology generally appears sound. My only significant concern is with the limited discussion of the potential for the stationary setup and small volume sampled by the receiver to underestimate the amount of air displacement present in more realistic conditions and, in turn, overestimate the efficacy of the PPE, particularly the face shield. Additionally, the English usage could be improved at times and the notation should be adjusted to match MDPI guidelines (specific instances given below). Once these details are added and the comments below have been addressed, I can confidently recommend publication in IJERPH.
SPECIFIC COMMENTS:
(1) – LN 65: Acronym "CDC" is not defined.
(2) – LN 75+: The numerals and units (e.g., "μm" on this line) should be separated by a space here and throughout the entire text.
(3) – LN 82: In addition to the floor surface area, it would be helpfull to provide the ceiling height and/or volume of the room.
(4) – LN 110: Sentence should start as "Two types of personal...".
(5) – LN 112: The "Spanish brand" should be specified.
(6) – LN 132: There is information here that was already provided on page 3 (e.g., airflow speed, size channels). The redundancies should be corrected, and I would recommend restructuring so that the other related content on page 3 appears closer to or inside of this paragraph.
(7) – LN 137: 1.416 litters strikes me as a very small sample volume, and one that is comparable to the amount of space behind the face shield. I worry that the significant reduction factor found with the face shield on the receiver is simply caused by an insufficient amount of air being displaced for the emitted particles to enter the particle detector. Said another way, the space behind the face shield may be serving as a kind of reservoir of clean air for the receiver. If the shield were worn in a more realistic context by a human, motion and extended respiration (1.416 litters equates to only around two breaths) would likely quickly display the original "clean" air behind the mask. The potential impact of more realistic, prolonged exposure in which meaningful air displacement occurs needs to be discussed and statements regarding the efficacy of the receiver face shield should be softened accordingly.
(8) – LN 149: These size channels have already been defined, their presence here is redundant.
(9) – LN 154: As I understand it, "particles emitted" actually represents the amount of particles received without any PPE, not the total number of particles released by the emitter. If this is the case, another more appropriate term should be used.
(10) – LN 168: The role of droplet evaporation under these temperature and humidity conditions should be discussed. Ideally, calculations of droplet evaporation rate as a function of particle size would be provided here or at another point in the manuscript.
(11) – LN 169: The exact variable (presumably RH?) that had a value 68.3% should be clarified.
(12) – LN 173: Should read "...coming out of the pipe...".
(13) – LN 177: It might be better to specify the number of particles per unit volume so that the reader can better contextualize these numbers with other works.
(14) – Table 1/Figure 4: The style of this table and figure should be made in line with MDPI guidelines (e.g., period as decimal separator, scientific notation when appropriate, etc.).
(15) – LN 192: This paragraph needs rewording for improved clarity. It would help to clarify that there where two cases with RF=96.7%.
(16) – Table 2: This is basically just an abridged version of Table 1. The only new information are the two p-values, which are already stated in the text. I would recommend eliminating this table.
(17) – LN 235: The fact that these experiments correspond to intense, but very short, exposure needs to be emphasized here.
(18) – LN 249: I'm not sure "manuporting" is the correct word, I would advise the authors to rephrase.
(19) – LN 257: I think the authors mean to say "peak" instead of "pic".
(20) – LN 313: The word "interesting" is not appropriate. "Desirable" might be a better choice.
(21) – LN 329: The word "parameter" is not appropriate here. "Consideration" might be a better choice.
(22) – LN 353: I'm not sure what is meant by "...mist on the corrective glasses...". Are the authors perhaps referring to fog on eyeglasses? If so, why would the face shield prevent this phenomenon?
(23) – LN 362: The p/s values in parenthesis are quite small, and the authors of the cited study from which the values are derived acknowledge significant limitations in their methodology that would result in an undercounting of particles (not fully sampling particles smaller than 0.5 μm, incomplete capture of exhaled air volume, etc.). In fact, these values underestimate more complete results in others studies by multiple orders of magnitude (e.g., Morawska et al., 2009; Schwarz et al., 2010). Either the caveats of Asadi et al. (2020) need to be noted, or other, more complete, values need to be provided.
(24) – LN 388: I believe the term "school cafeterias" may be more appropriate than "school canteens".
REFERENCES:
Schwarz, Katharina, et al. "Characterization of exhaled particles from the healthy human lung—a systematic analysis in relation to pulmonary function variables." Journal of aerosol medicine and pulmonary drug delivery 23.6 (2010): 371-379.
Morawska, L. J. G. R., et al. "Size distribution and sites of origin of droplets expelled from the human respiratory tract during expiratory activities." Journal of Aerosol Science 40.3 (2009): 256-269.
Author Response
The authors present results from an experiment aiming to quantify the ability of different forms of Personal Protective Equipment (PPE) to limit aerosol transmission between two individuals. The two pieces of PPE studied include face shields and masks. The experiments conducted aim to reproduce quick, but proximal and intense, periods of exchange between two individuals, similar to situations common in healthcare settings. The authors conclude that both masks and face shields are very effective at reducing emitted particles, while the face shield provided moderate protection to the receiving individual but more than masks which provided only very limited protection when worn by the receiver.
The work is obviously quite relevant given the present CoVID-19 pandemic and the subject matter is certainly appropriate for IJERPH. The manuscript is well organized and clear and the methodology generally appears sound.
Dear Reviewer,
Thank you for appreciating our manuscript.
My only significant concern is with the limited discussion of the potential for the stationary setup and small volume sampled by the receiver to underestimate the amount of air displacement present in more realistic conditions and, in turn, overestimate the efficacy of the PPE, particularly the face shield.
We clarified and better insisted on this point under discussion.
Additionally, the English usage could be improved at times and the notation should be adjusted to match MDPI guidelines (specific instances given below). Once these details are added and the comments below have been addressed, I can confidently recommend publication in IJERPH.
Thank you very much for your recommendation and for the specific comments below.
SPECIFIC COMMENTS:
(1) – LN 65: Acronym « CDC » is not defined.
It's added.
(2) – LN 75+: The numerals and units (e.g., "μm" on this line) should be separated by a space here and throughout the entire text.
We have made these changes throughout the text.
(3) – LN 82: In addition to the floor surface area, it would be helpfull to provide the ceiling height and/or volume of the room.
We added it in the text.
(4) – LN 110: Sentence should start as "Two types of personal...".
It is modified.
(5) – LN 112: The "Spanish brand" should be specified.
We have made this clear.
(6) – LN 132: There is information here that was already provided on page 3 (e.g., airflow speed, size channels). The redundancies should be corrected, and I would recommend restructuring so that the other related content on page 3 appears closer to or inside of this paragraph.
Indeed, we have moved these elements from page 3 to page 5 to limit redundancy.
(7) – LN 137: 1.416 litters strikes me as a very small sample volume, and one that is comparable to the amount of space behind the face shield. I worry that the significant reduction factor found with the face shield on the receiver is simply caused by an insufficient amount of air being displaced for the emitted particles to enter the particle detector. Said another way, the space behind the face shield may be serving as a kind of reservoir of clean air for the receiver. If the shield were worn in a more realistic context by a human, motion and extended respiration (1.416 litters equates to only around two breaths) would likely quickly display the original "clean" air behind the mask. The potential impact of more realistic, prolonged exposure in which meaningful air displacement occurs needs to be discussed and statements regarding the efficacy of the receiver face shield should be softened accordingly.
We detailed all this in the discussion.
(8) – LN 149: These size channels have already been defined, their presence here is redundant.
We have removed this repetition.
(9) – LN 154: As I understand it, "particles emitted" actually represents the amount of particles received without any PPE, not the total number of particles released by the emitter. If this is the case, another more appropriate term should be used.
You are right, we had simplified to "emitted particles", but for the sake of brevity, we have replaced "received particles without EPP" in the equation.
(10) – LN 168: The role of droplet evaporation under these temperature and humidity conditions should be discussed. Ideally, calculations of droplet evaporation rate as a function of particle size would be provided here or at another point in the manuscript.
We better discussed this point.
(11) – LN 169: The exact variable (presumably RH?) that had a value 68.3% should be clarified.
Indeed, it was the relative humidity, it was added.
(12) – LN 173: Should read "...coming out of the pipe...".
Fixed.
(13) – LN 177: It might be better to specify the number of particles per unit volume so that the reader can better contextualize these numbers with other works.
We have added this value.
(14) – Table 1/Figure 4: The style of this table and figure should be made in line with MDPI guidelines (e.g., period as decimal separator, scientific notation when appropriate, etc.).
We have corrected these errors.
(15) – LN 192: This paragraph needs rewording for improved clarity. It would help to clarify that there where two cases with RF=96.7%.
We have clarified the wording of this paragraph.
(16) – Table 2: This is basically just an abridged version of Table 1. The only new information are the two p-values, which are already stated in the text. I would recommend eliminating this table.
Indeed, we have therefore deleted this table.
(17) – LN 235: The fact that these experiments correspond to intense, but very short, exposure needs to be emphasized here.
We have added a sentence to add the main features of our experiment here.
(18) – LN 249: I'm not sure "manuporting" is the correct word, I would advise the authors to rephrase.
Right, it was an anglicized verb...
(19) – LN 257: I think the authors mean to say "peak" instead of "pic".
(20) – LN 313: The word "interesting" is not appropriate. "Desirable" might be a better choice.
(21) – LN 329: The word "parameter" is not appropriate here. "Consideration" might be a better choice.
Thank you for these 3 corrections.
(22) – LN 353: I'm not sure what is meant by "...mist on the corrective glasses...". Are the authors perhaps referring to fog on eyeglasses? If so, why would the face shield prevent this phenomenon?
Indeed, we referred to « fog on eyeglasses ». The face shield limits the phenomenon compared to masks, since the hot air is better evacuated than under a mask.
(23) – LN 362: The p/s values in parenthesis are quite small, and the authors of the cited study from which the values are derived acknowledge significant limitations in their methodology that would result in an undercounting of particles (not fully sampling particles smaller than 0.5 μm, incomplete capture of exhaled air volume, etc.). In fact, these values underestimate more complete results in others studies by multiple orders of magnitude (e.g., Morawska et al., 2009; Schwarz et al., 2010). Either the caveats of Asadi et al. (2020) need to be noted, or other, more complete, values need to be provided.
We have taken over the more complete data from the cited studies, thank you for the references.
(24) – LN 388: I believe the term "school cafeterias" may be more appropriate than "school canteens".
It is modified, thank you.
We thank you again for your opinion and your precious remarks.
Reviewer 2 Report
Wendling et al. investigated the effect of a face mask and a face shield on the transmission of aerosols to conclude that a face shield is more efficient in removing particles when two people are close (25 cm). The study is of great interest to the public, and the experiment is well designed. I suggest for publication after considering the following points:
1. When showing the aerosol numbers, please indicate the unit. Are they all in # cm-3? Table 1, 2, and A1, Figure 4, Section 3.2.
2. Figure 4: The orange line is not "amount of accumulated particles".
3. Shield has a higher RF could be because the distance is very close. The particle inertia effect is more significant, especially for the case where the Em is wearing a shield.
4. Line 181: 0.03 -> 0.3?
5. What kind of liquid was used to generate aerosols? Is it a solution or pure water?
6. Figure 5 is confusing. The bar plot is not consistent with the data in the photos.
7. Figure 6: please correct the symbols.
8. How is the background aerosol level changed after the experiment is done?
9. Figure 7: The concept of "PIP with Em" is confusing. It is still an evaluation from the Re position, right? But the value is significantly lower than that of "PIP with Re". Also, how does background level affect this result? If it is for Re, the background should be 100% for Re since Re does not have any PPE.
10. The concept of double protection is more likely to wear both a face mask and a shield for an individual. What is the effect of that combination?
11. To conclude that a face shield performs better than a face mask, I would also test the case where the Re and the Em are at least 6 feet separated. Otherwise, the conclusion can only apply for close contact.
Author Response
Wendling et al. investigated the effect of a face mask and a face shield on the transmission of aerosols to conclude that a face shield is more efficient in removing particles when two people are close (25 cm). The study is of great interest to the public, and the experiment is well designed. I suggest for publication after considering the following points:
Dear reviewer,
Thank you very much for your interest in our work.
- When showing the aerosol numbers, please indicate the unit. Are they all in # cm-3? Table 1, 2, and A1, Figure 4, Section 3.2.
We presented the absolute values (mean number of total particles received by the receptor per 1.416 liters). We have added one variable per unit, for a better comparison with the literature.
- Figure 4: The orange line is not "amount of accumulated particles".
Indeed, we fixed it.
- Shield has a higher RF could be because the distance is very close. The particle inertia effect is more significant, especially for the case where the Em is wearing a shield.
We add this pertinent sentence in our discussion.
- Line 181: 0.03 -> 0.3?
Fixed, thank you.
- What kind of liquid was used to generate aerosols? Is it a solution or pure water?
We used pure water; we add this in methods section.
- Figure 5 is confusing. The bar plot is not consistent with the data in the photos.
- Figure 6: please correct the symbols.
We simplified the 2 figures and corrected the symbols.
- How is the background aerosol level changed after the experiment is done?
The background aerosol level was checked between each experiment and was between 11,383 and 17,955 particles. We have made this clearer in the text
- Figure 7: The concept of "PIP with Em" is confusing. It is still an evaluation from the Re position, right? But the value is significantly lower than that of "PIP with Re". Also, how does background level affect this result? If it is for Re, the background should be 100% for Re since Re does not have any PPE.
We have clarified the title and the figure. The involvement of the background aerosol level is more important for the larger particles. Results are also available in Annex.
- The concept of double protection is more likely to wear both a face mask and a shield for an individual. What is the effect of that combination?
Unfortunately we have not studied this configuration. We add it as a limitation in our discussion and clarify the term "double protection" in our manuscript.
- To conclude that a face shield performs better than a face mask, I would also test the case where the Re and the Em are at least 6 feet separated. Otherwise, the conclusion can only apply for close contact.
We have reformulated our conclusions in line with the remarks of the other reviewers and yours.
Thank you for your interest in our work.
Reviewer 3 Report
The aim of the article submitted by Wendling et al. was to evaluate the comparative barrier performance effect of two different types of protection (mask and face shield) during different situations. The article is quite clear and interesting; the paper is well-written. However, I have a few points that should be revised.
These points include:
- Materials and Methods could be differently organized for better understanding. My suggestion is to divide this paragraph into three different subsections, if it is possible: 1) experimental setup including the description of the heads and masks (see lines 80-114); 2) description and analysis of different test setups ( lines 117-164); 3) statistical analysis
- Line 81: the reason why you decided to put heads at 25cm from each other should be also explained in this paragraph
- Figure 6: p-value in range 2,5-5u does not correspond with the p-value written in the text (line 207).
- Appendix tables: there is one table (Table 1) at page 14 including all results of aerosol particle measurements, but in “Results” you wrote Appendix table 3 (line 223), and it should be corrected.
- Line 293 “ve” should be corrected with "we"
Author Response
The aim of the article submitted by Wendling et al. was to evaluate the comparative barrier performance effect of two different types of protection (mask and face shield) during different situations. The article is quite clear and interesting; the paper is well-written. However, I have a few points that should be revised.
Dear Reviewer,
Thank you for your interest and appreciation of our work.
These points include:
- Materials and Methods could be differently organized for better understanding. My suggestion is to divide this paragraph into three different subsections, if it is possible: 1) experimental setup including the description of the heads and masks (see lines 80-114); 2) description and analysis of different test setups ( lines 117-164); 3) statistical analysis
Thank you for the proposal we followed to clarify the reading of this section.
- Line 81: the reason why you decided to put heads at 25cm from each other should be also explained in this paragraph
We have explained our choice in this paragraph.
- Figure 6: p-value in range 2,5-5u does not correspond with the p-value written in the text (line 207).
Thank you, in agreement with the 2 other reviewers, we have simplified the figures and left the p-values only in the text and/or in the appended table, to avoid redundancies.
- Appendix tables: there is one table (Table 1) at page 14 including all results of aerosol particle measurements, but in “Results” you wrote Appendix table 3 (line 223), and it should be corrected.
- Line 293 “ve” should be corrected with "we"
Thank you, we have corrected these 2 items.
Thank you for your interest in our work.
This manuscript is a resubmission of an earlier submission. The following is a list of the peer review reports and author responses from that submission.